# Pathogenesis of Respiratory Syncytial Virus Infection in BALB/c Mice Differs Between Intratracheal and Intranasal Inoculation

**DOI:** 10.3390/v11060508

**Published:** 2019-06-03

**Authors:** Elisabeth A. van Erp, Anke J. Lakerveld, H. Lie Mulder, Willem Luytjes, Gerben Ferwerda, Puck B. van Kasteren

**Affiliations:** 1Centre for Infectious Disease Control, National Institute for Public Health and the Environment (RIVM), 3721 MA Bilthoven, The Netherlands; liz.van.erp@rivm.nl (E.A.v.E.); anke.lakerveld@rivm.nl (A.J.L.); mulder@viroclinics.com (H.L.M.); willem.luytjes@rivm.nl (W.L.); 2Section Pediatric Infectious Diseases, Laboratory of Medical Immunology, Radboud Institute for Molecular Life Sciences, Radboudumc, 6525 GA Nijmegen, The Netherlands; gerben.ferwerda@radboudumc.nl; 3Radboud Center for Infectious Diseases, Radboudumc, 6525 GA Nijmegen, The Netherlands

**Keywords:** RSV, animal model, pathology, anesthesia, inoculation method, inflammation

## Abstract

Human respiratory syncytial virus (RSV) is a major cause of severe lower respiratory tract disease requiring hospitalization in infants. There are no market-approved vaccines or antiviral agents available, but a growing number of vaccines and therapeutics are in (pre)clinical stages of development. Reliable animal models are crucial to evaluate new vaccine concepts, but in vivo RSV research is hampered by the lack of well-characterized animal models that faithfully mimic the pathogenesis of RSV infection in humans. Mice are frequently used in RSV infection and vaccination studies. However, differences in the use of mouse strains, RSV subtypes, and methodology often lead to divergent study outcomes. To our knowledge, a comparison between different RSV inoculation methods in mice has not been described in the literature, even though multiple methods are being used across different studies. In this study, we evaluated various pathological and immunological parameters in BALB/c mice after intratracheal or intranasal inoculation with RSV-A2. Our study reveals that intranasal inoculation induces robust pathology and inflammation, whereas this is not the case for intratracheal inoculation. As immunopathology is an important characteristic of RSV disease in infants, these data suggest that in mice intranasal inoculation is a more appropriate method to study RSV infection than intratracheal inoculation. These findings will contribute to the rational experimental design of future in vivo RSV experiments.

## 1. Introduction

Respiratory syncytial virus (RSV) infection is a major cause of severe lower respiratory illness requiring hospitalization in young infants [1]. Currently, there are no market-approved vaccines available, although multiple vaccines are in (pre)clinical stages of development (https://www.path.org/resources/rsv-vaccine-and-mab-snapshot/). Reliable animal models are crucial to evaluate new vaccine concepts. Several animal models have been developed to study RSV infection and disease, including mice, cotton rats, chimpanzees, cattle, and sheep [2,3]. However, none of these models fully replicates the pathogenesis of RSV infection in humans. In light of this, it is pivotal to obtain a thorough understanding of the implications of the use of different methodologies, in order to design experimental approaches that are most suitable for answering specific research questions.

Despite their limitations, mice are the most commonly used animal species for in vivo modeling of RSV disease. Most inbred mouse strains, including the widely used BALB/c and C57 Bl/6 mice, are semi-permissive for RSV infection. These mice need an inoculum with a high viral titer to detect any lower respiratory tract disease symptoms and general measures of illness such as weight loss [4,5]. In addition, the innate and adaptive immune responses after RSV infection differ between humans and mice. For example, mice show a limited recruitment of neutrophils to the lungs, whereas neutrophils are the most abundant cell type during human RSV infection [4,6]. However, the availability of a vast array of mouse-specific reagents and molecular tools makes mice indispensable for gaining mechanistic insights into RSV infection and disease.

Not only is the translation from mouse to human difficult for RSV infection, the comparison between mouse studies is also challenging. This is due in part to differences arising from the use of varying mouse strains and RSV subtypes. The mouse strain is a major determinant of RSV susceptibility and subsequent pathogenesis [7,8]. In addition, responses vary greatly between different RSV subtypes and (clinical) isolates, even if the same mouse strain is used [9,10]. It has also been found that the cell line used for RSV propagation and the purity of the viral inoculum influence infectivity and pathology [11,12,13]. These reports highlight that appropriate selection of the study setup is crucial to allow for proper interpretation of the results.

For multiple pathogens other than RSV, it has been suggested that different infection routes also play a role in the inconsistent outcomes between studies. For example, both the site of pathogenesis as well as mortality rates after influenza virus infection in ferrets differed between intranasal and intratracheal inoculation [14,15]. Moreover, different outcomes were observed for intranasal versus intratracheal infection of mice with Listeria monocytogenes [16] and *Pseudomonas aeruginosa* [17]. To our knowledge, comparison between these inoculation methods, with regard to viral load, pathogenesis, and the immune response has never been undertaken for RSV infection in mice. Although intranasal inoculation methods are most common within the RSV field [18], intratracheal inoculation has also been described [19,20,21].

In this study, we evaluated virus-induced pathology and the antiviral immune response in BALB/c mice after intratracheal or intranasal inoculation with RSV-A2. Although mice in both groups received an equal dose of the same virus stock, striking differences in viral load, lung damage, and inflammatory mediators were apparent. Our study reveals that at three days post-infection, intranasal inoculation results in robust pathology and immune activation, whereas intratracheal inoculation barely induces either lung damage or inflammation. As immunopathology is an important characteristic of RSV disease in infants, these data suggest that intranasal inoculation in mice is a more appropriate method to mimic the effect of RSV infection in humans than intratracheal inoculation. Together, these findings provide important insights that are essential for the rational design of future in vivo experiments with RSV.

## 2. Materials and Methods

### 2.1. Mice

Female, specific-pathogen-free BALB/c mice (7–9 weeks old) were obtained from Charles River Laboratories and kept at the animal facilities of Intravacc (Netherlands). The animals were allowed to acclimatize for at least one week. Different batches of mice were used for all five experiments. The study was approved by the Animal Ethical Committee of the Netherlands (CCD; 20185186; April 23, 2018). All experiments were performed in accordance with the guidelines of the institutional animal care committee.

### 2.2. Cells and Virus

HEp-2 cells (ATCC CCL-23) were cultured in minimum essential medium (MEM; Gibco, Thermo Fisher Scientific, Waltham, MA, USA) supplemented with 10% heat-inactivated fetal calf serum, and 1% penicillin/streptomycin/glycamin (Gibco). Human RSV-A2 (ATCC, VR1302) was propagated in HEp-2 cells. The virus stock was purified between layers of 10% and 50% sucrose by ultracentrifugation. Virus titer was determined by 50% tissue culture infective dose (TCID50) assay on HEp2 cells. TCID50/ml was calculated using the Spearman and Karber method [22] and converted to plaque-forming units (pfu) per mL by multiplying by 0.69.

### 2.3. Inoculation of Mice

Two types of experiments were performed, intratracheal inoculation and intranasal inoculation. The intratracheal inoculation experiment was performed three times, whereas the intranasal inoculation experiment was performed twice. All experiments were carried out on different days. Every experiment consisted of two groups, one RSV- and one mock-infected, of three animals each. The different experiments are indicated by distinct symbols (shape and color) in the graphs.

On day 0, mice were intratracheally or intranasally infected with live, sucrose-purified RSV-A2. Preceding intratracheal inoculation, mice received an intraperitoneal injection of ketamine/xylazine (80 and 10 mg/kg, respectively). Mice were subsequently infected via a cannula in the trachea with 2 × 10^6^ pfu RSV-A2 diluted in phosphate buffered saline (PBS) in a total volume of 50 µL. Before intranasal inoculation, mice received anesthesia through inhalation of isoflurane (3.5% in O_2_). Mice were then infected with 2 × 10^6^ pfu RSV-A2 diluted in PBS in a total volume of 50 µL, which was administered dropwise to the nose. A schematic representation of the inoculation techniques is shown in Figure 1A. Control animals were mock inoculated either intratracheally or intranasally with 50 µL PBS containing 10% sucrose.

### 2.4. Sample Collection

Three days post-infection (dpi), mice were euthanized by exsanguination under isoflurane anesthesia. Bronchoalveolar lavage (BAL) samples and lungs were collected and processed as described below. Weights were measured daily during the course of the experiment.

BAL samples were obtained by infusing 0.8 mL or 0.9 mL (depending on weight of the animal) PBS into the lungs via the trachea, followed by aspiration into a syringe. The BAL sample was centrifuged at 500× *g* for 5 min. Cell pellets were collected for flow cytometric analysis and the supernatant was snap-frozen and stored at −80 °C for subsequent analysis of cytokine/chemokine and albumin concentrations.

Single-cell suspensions were prepared from mouse lungs by digestion with collagenase and DNase (Sigma-Aldrich, Saint Louis, MO, USA). Cells were pushed through 70 µm cell strainers and erythrocytes were lysed with ammonium-chloride-potassium (ACK) lysis buffer (0.155 M NH_4_CL, 10 mM KHCO_3_, and 0.1 mM Na_2_EDTA). Lung cells were used for determination of the viral load.

### 2.5. Viral Load

The amount of viral RNA was used as a proxy for viral load and was determined by quantitative reverse transcriptase PCR (RT-qPCR). Total RNA was extracted from lung single-cell suspensions using a Nucleospin RNA extraction kit (Machery-Nagel, Düren, Germany) and quantified on a Qubit fluorimeter (Invitrogen, Thermo Fisher Scientific). Subsequently, cDNA was synthesized using an iScript cDNA synthesis kit (Bio-Rad, Hercules, CA, USA) on a thermal cycler (ABI GeneAmp 9700, Applied Biosystems, Thermo Fisher Scientific). Finally, qPCR was performed on a real-time PCR machine (Applied Biosystems StepOne Plus) using SYBR Green reagents (Bio-Rad) and RSV-specific primers. Resulting C_T_ values were converted to arbitrary units (AU) using a standard curve. Amplification with primers corresponding to mouse actin mRNA was performed in parallel and used for normalization. Primer sequences are detailed in Table 1.

### 2.6. Flow Cytometric Analysis of Immune Cell Subsets in the BAL

BAL cells were isolated as described above and incubated with anti-mouse CD16/CD32 Fc receptor block (BD Pharmingen, BD Biosciences, San Jose, CA, USA) at 1 µg per 2 × 10^6^ cells for 10 min at 4 °C. Cells were subsequently incubated with the following antibodies: CD3-Pacific Blue (17A2, Biolegend, San Diego, CA, USA), CD4-APC (RM4-5, BD Pharmingen), CD8a-PE (53-6.7, Biolegend, Biolegend), CD11b-BV711 (M1/70, Biolegend), CD19-BV510 (1D3, BD Horizon, BD Biosciences), CD41-PE-CY7 (MWReg30, Biolegend), CD45-BUV395 (30-F11, BD Horizon), NKp46-BV605 (29A1.4, Biolegend), and Ly-6G-PE/Dazzle (1AB, Biolegend). Ultimately, cells were fixed in 3.7% paraformaldehyde in PBS for 10 min before flow cytometric analysis using an LSR Fortessa ×20 (BD Biosciences). FlowJo software V10 (FlowJo, BD, Franklin Lakes, NJ, USA) was used for data analysis. The gating strategy for one representative BAL sample is depicted in Appendix A.

### 2.7. Albumin and Cytokine/Chemokine Determination in BAL Fluid

Albumin concentrations were determined using a mouse albumin ELISA (ICL, Portland, OR, USA), according to the manufacturer’s protocol. Cytokine/chemokine concentrations in BAL samples were analyzed using the mouse anti-virus response panel LEGENDplex (Biolegend), according to the manufacturer’s protocol. This assay contained the following cytokines/chemokines: CCL2 (MCP-1), CCL5 (RANTES), CXCL1, CXCL10 (IP-10), IFN-α, IFN-β, IFN-γ, IL-1β, IL-6, IL-10, IL-12, GM-CSF, and TNF.

### 2.8. Statistical Methods

All statistical analyses were performed with Prism 7 (GraphPad Software, San Diego, CA, USA). Viral load, albumin concentrations, cell counts, and cytokine/chemokine concentrations were log transformed before statistical testing. Comparisons between two groups were performed using an unpaired Student’s t-test. Multiple comparisons were analyzed using a one-way analysis of variance (ANOVA), followed by Tukey’s multiple comparisons test. Differences in mouse body weight between mock- and RSV-inoculated animals were analyzed using a two-way ANOVA, using the day and group as variables, followed by Sidak’s multiple comparisons test. *p* values <0.05 were considered statistically significant.

## 3. Results

### 3.1. Intranasal Inoculation Results in Slightly Higher Viral Load and More Pronounced Pathology Compared to Intratracheal Inoculation

To compare the effect of the inoculation method on viral load and pathology, BALB/c mice were (mock) infected with vehicle control or RSV-A2 either intratracheally (*n* = 9 per group, divided over three experiments) or intranasally (*n* = 6 per group, divided over two experiments). We pooled all data per group for subsequent analysis. Separate experiments are indicated in the graphs by distinct shapes and colors of symbols. The technical differences between intratracheal and intranasal inoculation are illustrated in Figure 1A.

The viral load was determined at three dpi by assessing the amount of viral RNA in total lung cells using RT-qPCR. The results showed a higher mean lung viral load in intranasally inoculated compared to intratracheally inoculated animals (Figure 1B). Whereas all intranasally infected animals tested positive for viral RNA in the lungs, two animals that had been intratracheally inoculated tested negative for viral RNA. This suggests that intratracheal inoculation is more error-prone than intranasal inoculation, which is likely due to the possibility of inadvertent administration of the inoculum into the esophagus. The two RSV-negative animals were excluded from all further analysis. Upon exclusion of these two animals, the mean viral load was approximately 2-fold higher in the intranasally compared to the intratracheally inoculated animals, which was a statistically significant difference (Figure 1C). No virus was detected in any of the mock-inoculated animals.

To determine whether the high viral load upon intranasal inoculation was accompanied by more pronounced pathology in the lung, we subsequently quantified the albumin concentration in the bronchoalveolar lavage (BAL) fluid and assessed changes in body weight during the course of the experiment. Leakage of serum albumin from the circulation into the lung lumen can be used as a measure for lung damage [23]. Although intratracheal RSV inoculation did not result in any observable lung damage, with albumin concentrations comparable to those of mock inoculation, intranasal inoculation with RSV resulted in higher albumin concentrations compared to both intranasal mock inoculation and intratracheal RSV inoculation, although this was not statistically significant (Figure 1D). In addition to lung damage, animals that had received RSV through intranasal inoculation showed a decrease in body weight, losing almost 5% of body weight at day 1 (Figure 1E). In contrast, animals that had received an intratracheal inoculation did not show a decrease in body weight, but rather gained weight throughout the experiment. Intranasally mock-inoculated animals did not show a change in body weight.

### 3.2. Virus-Induced Pulmonary Cellular Influx is More Pronounced Upon Intranasal Compared to Intratracheal Inoculation

To investigate the effect of inoculation method on inflammatory parameters, we assessed pulmonary cellular influx by determining (specific) immune cell counts in the BAL at three dpi by flow cytometry. The total leukocyte count in the BAL was significantly increased upon RSV infection compared to that of mock infection for both inoculation methods (Figure 2A). In addition, total leukocyte counts were significantly higher upon intranasal RSV inoculation compared to those of intratracheal RSV inoculation. Cell type-specific analysis revealed that natural killer (NK) cell, neutrophil, antigen-presenting cell (APC), and CD4 and CD8 T cell counts were increased upon RSV infection compared to those of mock infection for both inoculation methods (Figure 2B–F). These differences were statistically significant, except for neutrophils. In addition, NK cell, neutrophil, APC, and CD4 and CD8 T cell counts were markedly higher upon intranasal compared to intratracheal RSV inoculation. These differences were statistically significant, except for CD4 T cells. No increased B cell counts were detected after intratracheal or intranasal RSV inoculation in the BAL at three dpi. These data indicate that the observed pulmonary influx is composed of a variety of immune cells and does not appear to be linked to influx of one particular cell type. Overall, we found that at three dpi the pulmonary cellular influx in the BAL induced by intranasal RSV inoculation was much more pronounced than that induced by intratracheal RSV inoculation.

### 3.3. Elevated Virus-Induced Pulmonary Cellular Influx upon Intranasal Inoculation is Accompanied by Increased Cytokine and Chemokine Levels

The influx of immune cells in the lungs is a hallmark of inflammation and is often accompanied by an increase in the levels of (pro-inflammatory) cytokines and chemokines. To investigate whether intranasal inoculation results in elevated secretion of soluble pro-inflammatory mediators compared to intratracheal inoculation, we determined the concentration of various cytokines and chemokines in the BAL fluid at three dpi. We focused on soluble mediators that are known to be involved in the response against viral infection.

Compared to mock infection, RSV infection via both inoculation methods resulted in increased BAL concentrations of CCL2 (MCP-1), CCL5 (RANTES), CXCL1, CXCL10 (IP10), IFN-α, IFN-γ, and TNF (Figure 3A–G). These differences were statistically significant, except for CCL5 and IFN-α in the case of intratracheal inoculation. IL-1β, IL-12, and GM-CSF showed no difference upon RSV infection compared to mock infection and the levels of IFN-β, IL-6, and IL-10 remained below the limit of detection. Compared to intratracheal RSV inoculation, mice that had been intranasally inoculated with RSV showed 3- to 8-fold higher levels of CCL2, CCL5, CXCL1, CXCL10, IFN-α, IFN-γ, and TNF (Figure 3H). These differences were all statistically significant. Similar to the pulmonary cellular influx, we observed differences in some of the pro-inflammatory mediators between experiments performed on different days. In summary, RSV infection induced expression of a number of pro-inflammatory cytokines and chemokines. Animals that were inoculated intranasally showed markedly higher levels of RSV-induced pro-inflammatory mediators in BAL fluid than intratracheally inoculated animals at three dpi.

## 4. Discussion

Seemingly minor methodological differences can have profound effects on the outcome of animal studies. It is therefore of pivotal importance to understand beforehand the implications of choosing a particular method, in order to allow for proper interpretation of the results obtained at the end of the experiment. In the present study, we compared the pathological and immunological effects of RSV infection via intratracheal or intranasal inoculation. The latter resulted in a slightly higher viral load that was accompanied by more pronounced pathology compared to intratracheal inoculation. In addition, inflammatory parameters such as pulmonary cellular influx and pro-inflammatory cytokine and chemokine levels were significantly higher after intranasal versus intratracheal inoculation.

A possible explanation for the observed differences in viral load, pathology, and host responses is the difference in (primary) infection site between intratracheal and intranasal inoculation. During intratracheal inoculation, the virus is delivered directly to the lung, presumably resulting in an infection that is confined to the lower respiratory tract, at least initially. In contrast, intranasal inoculation with the volume used in this study results in infection of both the upper and lower respiratory tract [24]. It is conceivable that viral replication in the upper airways contributes to a higher viral load in the lungs. In addition, it is likely that the host response to viral infection in the upper airways influences the immune response in the lower airways. Previous studies have shown that viral replication and host responses may vary depending on the anatomical localization of the RSV infection. For example, in vitro studies have shown increased viral production in human nasal epithelial cells compared to bronchial epithelial cells [25]. In contrast, ex vivo studies with primary pediatric epithelial cell cultures have shown lower RSV titers accompanied by slightly lower cytokine responses in nasal cells compared to bronchial cells [26]. Interestingly, influenza virus infection in ferrets resulted in higher mortality rates after intratracheal inoculation compared to intranasal inoculation [15]. These findings highlight that virus-specific information on the outcomes of different inoculation routes is essential for the design of future studies.

Another important factor that might have contributed to the observed differences is the differential use of anesthetics between the two inoculation methods. For intratracheal inoculation, which is a more invasive procedure, the induction of anesthesia through ketamine/xylazine injection is common practice. Intranasal inoculation only requires a short duration of anesthesia and is therefore performed under inhaled isoflurane anesthesia. Although not specifically investigated for RSV infection, the use of anesthetics can have a major impact on the immune system [27,28,29,30]. Multiple studies have reported that ketamine possesses anti-inflammatory properties and impairs the release of pro-inflammatory cytokines [29,31,32]. In addition, ketamine has been shown to have antiviral properties against rabies virus [33]. Although isoflurane has also been shown to reduce inflammation, these effects were mostly seen after long-term exposure [30,34,35]. Taken together, the use of ketamine/xylazine anesthesia for intratracheal inoculation may, at least in part, explain the observed lower immunopathology compared to intranasal inoculation. Considering this, we found it striking that a considerable number of published studies fail to specify the type of anesthesia that was used during RSV infection in mice.

The main limitation of our study is the fact that the intratracheal and intranasal inoculations were performed in separate experiments, which is a suboptimal experimental setup to compare the two methods. However, during each experiment, exactly the same virus dose was used, originating from the same virus batch. In addition, all animals were obtained from the same source, were of similar age, and were handled at the same facility. Nevertheless, we did observe differences between experiments performed on different days, especially regarding the immunological read-outs. As we used female mice for our study, a possible explanation for these effects is the ovarian cycle of the different batches of animals. Multiple studies have shown differences in the immune response between the different ovarian cycle phases [36,37]. For future experiments, the use of male mice might be more appropriate. However, this phenomenon also highlights the importance of performing experiments on multiple days to control for these kinds of variations. Taken together, we believe the experimental conditions of our study were similar enough to warrant a comparison between the two inoculation methods, especially considering the fact that the observed differences are quite striking and may have implications for the choice of inoculation method for future animal experiments.

Another limitation of our study is the fact that most pathological and immunological parameters were only assessed at one time-point post infection. It is possible that the observed differences are due to variations in the kinetics of virus replication and host response depending on the site of initial infection and/or anesthesia. However, the absence of a change in body weight upon intratracheal inoculation suggests this is not the main cause underlying the observed effects. Notably, we only assessed the short-term consequences of viral infection on pathology and host response. It remains a possibility that long-term immune responses, such as humoral and cellular memory responses, are differently affected by the two inoculation methods. Nevertheless, when focusing on these short-term responses, intranasal inoculation appears to be the method of choice.

In conclusion, we have demonstrated striking differences between intratracheal and intranasal RSV inoculation with regards to pathological and immunological read-outs, which may be explained by differences in the (initial) site of infection or the differential use of anesthetics. Based on our findings, intranasal inoculation appears to be better suited to the study of RSV-induced immunopathology than intratracheal inoculation. Not only does intranasal inoculation more accurately mimic the natural route of infection, it is also a less invasive procedure, requires milder anesthetics, is less error-prone, and, as we have demonstrated, induces more pronounced pathology and immune responses than intratracheal inoculation. Although it remains to be investigated whether the route of administration or the anesthetic is the main factor underlying the observed differences, it is evident that the selection of inoculation method (even when both techniques deliver the virus to the lower respiratory tract) is of crucial importance to obtain the most appropriate model to mimic RSV disease in humans.

## Figures and Tables

**Figure 1 viruses-11-00508-f001:**
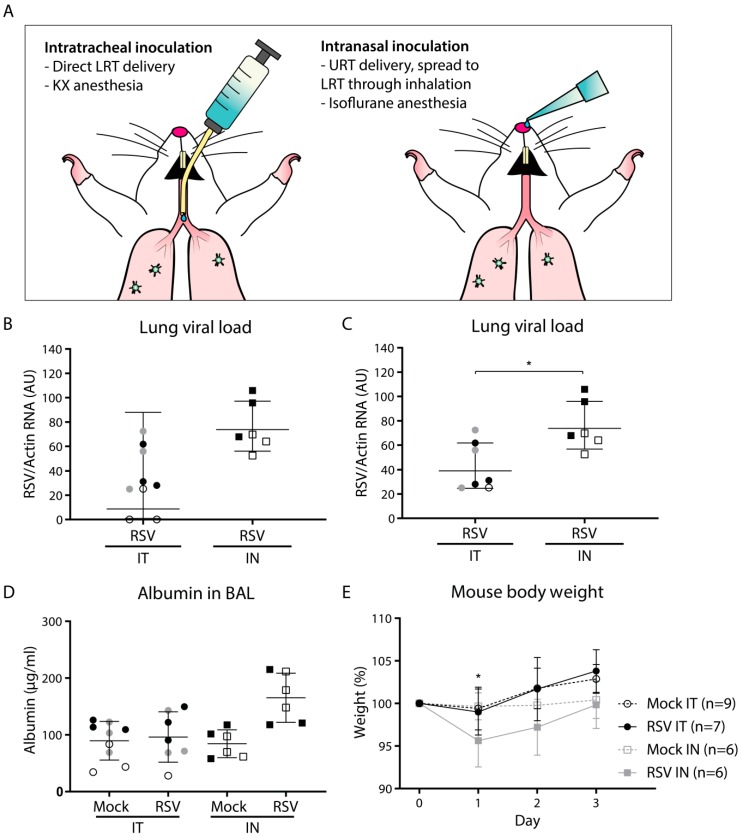
Viral load and pathology following intratracheal or intranasal respiratory syncytial virus (RSV) infection. Mice were (mock) inoculated either intratracheally or intranasally with 2 × 10^6^ pfu of live RSV-A2 or vehicle control. (**A**) Schematic figure of the intratracheal and intranasal inoculation methods. (**B**,**C**) Viral load in mouse lung cells as determined by RT-qPCR. Two animals that tested negative for RSV after intratracheal inoculation (depicted in panel B) were excluded from further analysis and removed from panel C. Geometric mean and SD are depicted. Viral load was log transformed before statistical comparison between the two groups was performed using an unpaired Student’s t-test (* *p* < 0.05). (**D**) Albumin concentration in the bronchoalveolar lavage (BAL) fluid measured by ELISA. Albumin concentrations were log transformed before analysis by a one-way ANOVA, followed by Tukey’s multiple comparisons test. Geometric mean and SD are depicted. (**E**) Mouse body weight with the weight at day 0 set to 100%. Each data point represents the mean weight (+/- SD) of all mice from the indicated group: mock intratracheal (IT) (*n* = 9), RSV IT (*n* = 7), mock intranasal (IN) (*n* = 6), RSV IN (*n* = 6). Body weights of mock- and RSV-infected animals were compared using a two-way ANOVA, followed by Sidak’s multiple comparisons test. Only the IN inoculation groups showed a difference between mock- and RSV-inoculated animals (* *p* < 0.05). Mice from separate experiments are indicated with a distinct symbol (IT: white/gray/black circles; IN: white/black squares). Abbreviations: AU, arbitrary units; BAL, bronchoalveolar lavage; IN, intranasal; IT, intratracheal; KX, ketamine-xylazine; LRT, lower respiratory tract; RSV, respiratory syncytial virus; SD, standard deviation; URT, upper respiratory tract.

**Figure 2 viruses-11-00508-f002:**
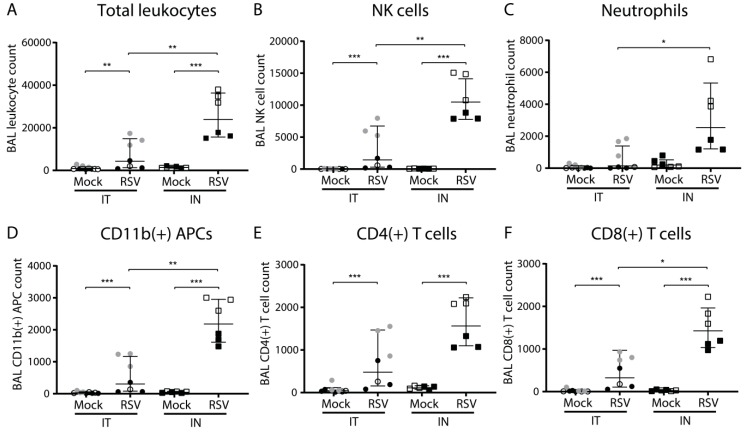
Pulmonary cellular influx following intratracheal or intranasal RSV infection. Mice were (mock) inoculated either intratracheally or intranasally with 2 × 10^6^ pfu of live RSV-A2 or vehicle control. Cells were isolated from the bronchoalveolar lavage (BAL) and analyzed using flow cytometry. Mice from individual experiments are indicated with a distinct symbol (IT: white/gray/black circles; IN: white/black squares). Graphs depict absolute cell counts for: (**A**) total leukocytes (CD45+), (**B**) NK cells (CD3-, CD56+), (**C**) neutrophils (CD3-, Ly-6G+, CD11b+), (**D**) antigen-presenting cells (CD3-, Ly-6G-, CD11b+), (**E**) CD4 T cells (CD3+, CD4+), and (**F**) CD8 T cells (CD3+, CD8+). All graphs depict geometric mean and SD. Cell count data were log transformed before analysis with a one-way ANOVA, followed by Tukey’s multiple comparisons test (* *p* < 0.05, ** *p* < 0.01, *** *p* < 0.001). Abbreviations: APCs, antigen-presenting cells; BAL, bronchoalveolar lavage; IN, intranasal; IT, intratracheal; NK cells, natural killer cells; RSV, respiratory syncytial virus; SD, standard deviation.

**Figure 3 viruses-11-00508-f003:**
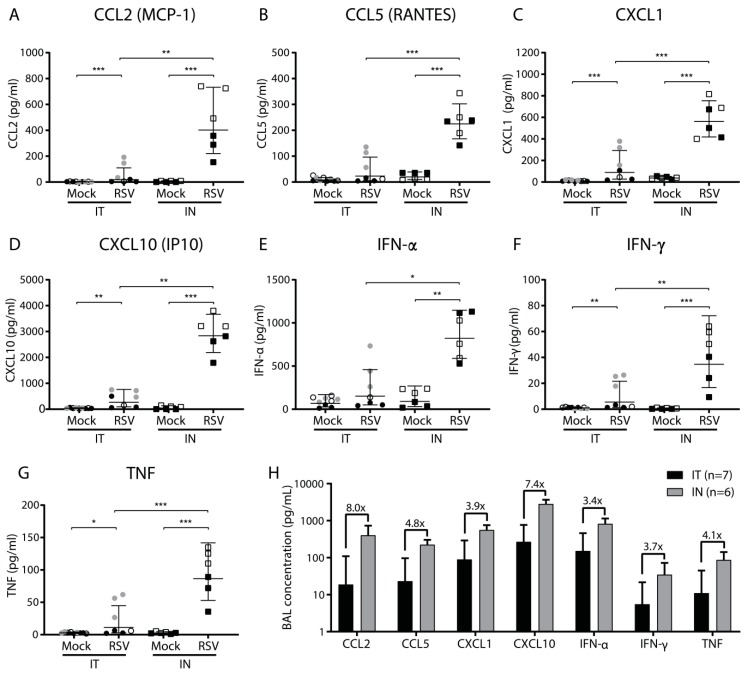
Cytokine and chemokine levels in the bronchoalveolar lavage following intratracheal or intranasal RSV infection. Mice were (mock) inoculated either intratracheally or intranasally with 2 × 10^6^ pfu of live RSV-A2 or vehicle control. Bronchoalveolar lavage (BAL) fluid was analyzed at three dpi using a multiplex mouse anti-virus response panel immunoassay. (**A**–**G**) Concentrations in pg/Ml of CCL2, CCL5, CXCL1, CXCL10, IFN-α, IFN-γ, and TNF. Mice from individual experiments are indicated with a distinct symbol (IT: white/gray/black circles; IN: white/black squares). Cytokine concentrations were log transformed before analysis with a one-way ANOVA, followed by Tukey’s multiple comparisons test (** *p* < 0.01, *** *p* < 0.001). (**H**) Concentration of indicated soluble pro-inflammatory mediators after either intratracheal (black bars) or intranasal (gray bars) RSV infection. The bars represent the geometric mean and SD of all animals belonging to the indicated groups. Fold change between intratracheal and intranasal infection is indicated above each pair of bars. Abbreviations: BAL, bronchoalveolar lavage; CCL, CC chemokine ligand; CXCL, CXC chemokine ligand; IN, intranasal; IP10, interferon gamma-induced protein 10; IT, intratracheal; MCP-1, monocyte chemoattractant protein 1; RANTES, Regulated upon Activation, Normal T cell Expressed and presumably Secreted; RSV, respiratory syncytial virus; SD, standard deviation.

**Table 1 viruses-11-00508-t001:** Primer sequences.

Transcript	Sequence
**RSV N**	Forward: 5′-TGACAGCAGAAGAACTAGAGGC-3′
	Reverse: 5′-TGGGTGATGTGAATTTGCCCT-3′
**Mouse Actin**	Forward: 5′-CGGTTCCGATGCCCTGAGGCTCTT-3′
	Reverse: 5′-CGTCACACTTCATGATGGAATTGA-3′

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
