# Peer review of "Pathogenesis of Respiratory Syncytial Virus Infection in BALB/c Mice Differs Between Intratracheal and Intranasal Inoculation"

_viruses, 2019, doi:10.3390/v11060508_

Reviewer 1 Report

A. van Erp et al. have submitted a primary research manuscript to Viruseson the differences in RSV pathogenesis observed in the Balb/c mouse model when the virus is delivered intranasally or via the trachea. This is a novel study that I feel is important in the field of RSV basic research. When I did an online search for ‘intratracheal’ ‘intranasal’ ‘RSV’ I could only find papers that used one or the other installation route. The authors identified an important distinction between the challenge routes in that their “study reveals that intranasal inoculation induces robust pathology and inflammation, whereas this is not the case for intratracheal inoculation.” Neutrophils and all other inflammatory markers that were tested were elevated in the intranasally instilled mice compared to all other treatment groups. There was the same pattern in the data when the authors looked at cytokines and chemokines. The authors have cited some of the influenza work that showed how influenza pathogenesis differs depending on the route of inoculation. Another key paper in this area was the Herfst paper published in Science (Science. 2012 Jun 22;336(6088):1534-41). This study demonstrated that ferrets succumbed to mammal adapted H5N1 during intratracheal instillation but not when the virus was administered intranasally. In the submitted manuscript, the experiments are well done and very well described. The paper is well written and thorough. 

Major criticism. The intra-trachea instilled mice were treated with ketamine xylazine whereas the intranasally instilled mice were not. The authors acknowledged this and even go so far as to criticize the volumes of published work on RSV that fails to specify the analgesia/anaesthesia that was used. The Herfst paper may help shed light on the discussion of this topic though the method of analgesia is not mentioned in this paper. This reviewer is not an author on Herfst et al. nor affiliated in any way.

Minor points. The fonts sizes in the figures are too small and need to be enlarged to be easily legible. 

Author Response

We thank the reviewer for the positive assessment of our manuscript and for appreciating its importance to the field. 

Concerning the reviewers first comment that different methods of anesthesia were used for both inoculation methods, we agree that this is suboptimal and prevents us to draw strong conclusions on the importance of the incoulation route per se. However, the use of standard operating procedures with respect to the type of anesthesia used for different inoculation methods made it impossible to set up the experiment differently. As the reviewer already indicates, we do extensively address this point in the discussion. Furthermore, we have added the suggested reference by Herfst et al. to the introduction and discussion section, which again shows the importance of virus-specific information on infection-induced pathology.

Concerning the reviewers second comment, we have enlarged the font sizes in all figures to allow for better readability.

We hope that we have addressed reviewer's concerns appropriately and would be happy to respond to any further questions regarding our manuscript.    

Reviewer 2 Report

The study by van Erp and co-workers addresses an important Problem in the study of respiratory Virus infections in General and RSV models in particular.

The current landscape of animal studies with These viruses makes use of different inoculation techniques, with intranasal and intratracheal methods being the mostly used ones. Thereby the author have shown that not only serious and statistically significant differences occur with respect to the immune reaction against respiratory viruses, but also that the more natural way, i.e. the intranasal infection results in more severe illness and immune reaction. This Major result is important and requires foremost Attention and has to be taken into account for all future studies in the field. 

Author Response

We thank this reviewer for the positive assessment of our manuscript and for recognizing the importance of our findings for the RSV field. 

There do not appear to be any comments to address.